# Recent Advances in Functionalization of Cotton Fabrics with Nanotechnology

**DOI:** 10.3390/polym14204273

**Published:** 2022-10-12

**Authors:** Tarek M. Abou Elmaaty, Hanan Elsisi, Ghada Elsayad, Hagar Elhadad, Maria Rosaria Plutino

**Affiliations:** 1Department of Textile Printing, Dyeing & Finishing, Faculty of Applied Arts, Damietta University, Damietta 34512, Egypt; 2Department of Spinning, Weaving and Knitting, Faculty of Applied Arts, Damietta University, Damietta 34512, Egypt; 3Istituto per lo Studio dei Materiali Nano Strutturati, ISMN—CNR, Palermo, c/o Department of ChiBio FarAm, University of Messina, Viale F. Stagno d’Alcontres 31, Vill. S. Agata, 98166 Messina, Italy

**Keywords:** multifunctional cotton fabrics, nanotechnology, metal nanoparticles

## Abstract

Nowadays, consumers understand that upgrading their traditional clothing can improve their lives. In a garment fabric, comfort and functional properties are the most important features that a wearer looks for. A variety of textile technologies are being developed to meet the needs of customers. In recent years, nanotechnology has become one of the most important areas of research. Nanotechnology’s unique and useful characteristics have led to its rapid expansion in the textile industry. In the production of high-performance textiles, various finishing, coating, and manufacturing techniques are used to produce fibers or fabrics with nano sized (10^−9^) particles. Humans have been utilizing cotton for thousands of years, and it accounts for around 34% of all fiber production worldwide. The clothing industry, home textile industry, and healthcare industry all use it extensively. Nanotechnology can enhance cotton fabrics’ properties, including antibacterial activity, self-cleaning, UV protection, etc. Research in the field of the functionalization of nanotechnology and their integration into cotton fabrics is presented in the present study.

## 1. Introduction

Textiles are commonly used in industries and households. The surface modification of textiles to impart multiple functions has recently gained a lot of attention. Researchers have successfully functionalized textiles for antibacterial, self-cleaning, flame retardant, UV protection, and enhanced performance properties (odor-fighting, anti-wrinkle, ant-pollen, and ant-static finishes [1]. Therefore, high-tech materials and fabric constructions will improve wearer comfort while incorporating distinctive features [2]. Among natural fibers, cotton is the most popular because of its softness, breathability, safety, low cost, regeneration performance, strength, elasticity, biodegradability, and hydrophilicity [3,4]. Cotton fabric does, however, have some disadvantages, including the possibility of microbial attacks on its fibrous structure, the ease with which creases form, and the loss of mechanical strength [5]. Microorganisms can easily grow and propagate on cotton fabrics because they are able to store humidity and have a high specific surface area [6]. A variety of fields, including health and medicine, have benefited from cotton fibers with antimicrobial properties [7]. Hygienic, functional, durable, and comfortable cotton fabrics are expected in modern times. Utilizing nanotechnology in cotton cloth is a significant challenge in achieving these characteristics and advancements [8]. Nanoparticles have been incorporated into textile finishing stages to address the inherent problems while also imparting functional properties to textile materials [9,10,11,12,13,14,15].

In a variety of applications, nanotechnology is widely regarded as having enormous potential around the world [16]. The textile industry has discovered nanotechnology, resulting in a new area of textile finishing called “Nano finishing”. Nano-sized particles have many desirable properties without adding a lot of weight, thickness, or stiffness to fabrics [17]. The first company to use nanotechnology in textiles was Nano-Tex, a subsidiary of Burlington Industries in the United States. As a result, a growing number of textile companies began investing in nanotechnology development [16]. While traditional textile finishing techniques do not always result in permanent effects and their functionality is lost after laundering or use, nanotechnology can provide a highly stable treatment [18,19].

In this review, we discuss recent developments in nanoparticles (primarily metals and metal oxide nanoparticles) used to modify and finish cotton fabrics from 2018 to 2022 to provide antimicrobial (antibacterial, antifungal), antiviral, UV protection, self-cleaning, water-repellent, and flame-retardant properties.

## 2. Common Types of Nanomaterials

There are many types of nanotechnology-produced materials, but the following four, in particular, are receiving significant attention:

### 2.1. Nanofinishing

The process of nanofinishing involves applying colloidal solutions or ultrafine dispersions of nanomaterials to fabrics in order to improve some of their functionalities [20]. In the case of nanofinishing, a smaller quantity of nanomaterials is required in comparison to the bulk materials used in traditional finishing achieving a similar effect. These nanofinishings do not alter the aesthetic feel of textile materials. They are more durable because they have a higher surface area-to-volume ratio in textile materials as well as a homogeneous distribution [21]. By using nanofinishing, existing processes can be improved, or new functional properties can be achieved that are not possible with traditional finishes [22].

### 2.2. Nanocoating

As part of nanocoating, a thin layer of less than 100 nm in thickness is deposited on a substrate to improve some properties or to add new functionality [23] such as enhanced color fastness, flame retardance, water or oil repellency, wrinkle resistance, and antimicrobial properties. Traditionally, textile coatings have thicknesses in the micrometer or millimeter range. However, conventional coatings can make fabrics completely impermeable, affecting their handling, feel, and breathability [24].

### 2.3. Nanofibers

As compared to conventional fibers, nanofibers have higher stiffness and tensile strength, as well as a very high surface area to weight ratio, low density, and a high pore volume. Because of these characteristics, nanofibers can be used in a wide variety of applications [25]. A variety of techniques can be used to fabricate nanofibers. One example of these techniques is phase separation, template synthesis, self-assembly fibers, and electrospinning (ELS). Electrospinning is a low-cost method for producing nanofibers [26].

### 2.4. Nanocomposites

It is possible to create nanocomposite fibers by dispersing nanosized fillers within a fiber matrix. Nanocomposite fibers can be developed with high electrical conductivity, superior strength, toughness, and lightweight using fillers such as nanosilicates, metal oxide nanoparticles, graphite nanofibers (GNFs), and single-wall and multi-wall carbon nanotubes (CNTs) [27].

## 3. Metal Nanoparticles (MNPs)

Among the nanomaterials used, metal nanoparticles (MNPs) are the most popular and versatile. For their diverse functional properties, numerous types of nanoparticles (NPs) have been integrated into various textile materials [28].

Inorganic nanoparticles, such as TiO_2_, ZnO, SiO_2_, Cu_2_O, CuO, Al_2_O_3_, and reduced graphene oxide, are more commonly used than organic nanoparticles because they can withstand high temperatures both thermally and chemically, their permanent stability under ultraviolet rays, and their non-toxicity [29,30]. A summary of the functions of metal nanoparticles can be found in Figure 1. Their ability to stick to fibers is also heavily influenced by their size. It is logical to assume that the largest particle cluster will easily be removed from the fiber surface, but the smallest particles will penetrate deeper and stick more firmly to the fabric. Reduced particle size results in changes in the material’s properties [31]. The presence of a reducing and stabilizing agent is essential in the preparation of these metallic nanoparticles. Metal nanoparticles are prepared by the reduction of metal salt solutions [32].

Nanoparticles are synthesized using a variety of physical, chemical, and biological methods [33,34]. The synthesis of NPs can be summarized in Figure 2. The nanoparticles synthesized using the green approach appear to be more stable and beneficial. In addition to being simple and cheap, it is also easy to characterize. A major advantage of green synthesis is that it produces nanoparticles with lower toxicity, making them less harmful to the environment [35,36].

### 3.1. Silver Nanoparticles(AgNPs)

Silver is one of the most popular antimicrobial nanoparticles. It acts as a doping antimicrobial agent and exhibits antimicrobial activity without affecting mechanical properties [37]. AgNPs have strong antiviral properties. Furthermore, AgNPs interactions with viruses can be improved by adjusting their physicochemical properties such as size, shape, surface charge, dispersion, and protein corona effects [38]. AgNPs may be applied to the surface of textile as part of a finishing process to functionalize them, such as spraying, or producing AgNPs directly on the surface of the fibre and inside it [39]. Cotton fabrics have been coated with AgNPs using a variety of techniques [40]. The functionalization of cotton fabrics incorporating AgNPs is summarized in Table 1.

Xu et al., 2018 [41] created durable antimicrobial cotton fabrics using AgNPs that were applied to cotton fabric using the pad-dry-cure technique. After 50 washing cycles, the cotton fabrics showed excellent antimicrobial activity (94%) against *Escherichia coli* and *Staphylococcus aureus*. Cotton’s original properties, such as tensile strength, water absorption, and vapor permeability, are not significantly affected by the modification. Rajaboopathi and Thambidura [42] fabricated functional cotton fabrics with AgNPs.

A seaweed extract (Padina gymnospora) was used to synthesize AgNPs, and citric acid was used as a crosslinker for applied AgNPs. The functionalized cotton fabrics were tested against *S. aureus* (Gram-positive) and *E coli* (Gram-negative). Cotton functionalized with AgNPs inhibited bacteria growth and provided better UV protection. A study by Patil et al. [43] used sonochemistry and deposition to create AgNPs-coated cotton fabrics with antimicrobial properties. They found that AgNPs uniformly deposited on cotton fabrics and showed excellent antibacterial activity against Gram-negative bacteria and Gram-positive bacteria. According to Ramezani et al., AgNPs produced by polyol methods were used to functionalize cotton fabrics with antibacterial and antifungal properties in 2019 [44]. A cotton textile coated with antimicrobial activity inhibited the growth of *S. aureus, E. coli*, and *Candida albicans*. In 2020, Maghimaa et al. [45] evaluated the antimicrobial and wound-healing activity of coated cotton fabric with AgNPs. Peltophorum pterocarpum leaf extracts were used in the synthesis of AgNPs. The AgNPs cotton fabrics showed a good zone of inhibition against *S. aureus*, *Pseudomonas aeruginosa, Streptococcus pyogenes,* and *C. albicans* and good wound healing activity when tested against fibroblast. The antibacterial activity of functionalized textiles with AgNPs against *E. coli*, *S. aureus*, *P. aeruginosa*, *Klebsiella pneumoniae*, *Klebsiella oxytoca*, and *Proteus mirabilis*, and antifungal activities against *Aspergillus niger* were reported by Aguda and Lateef [46]. AgNPs were synthesized using wastewater from fermented seeds of Parkia biglobosa. Using a pad-dry-cure approach, AgNPs were applied to cotton and silk. The AgNPs-functionalized textiles prevented bacteria growth up to the fifth cycle of washing. In the same year, Deeksha et al. [47] developed antibacterial cotton fabrics with AgNPs using the medicinal plant Vitex leaf extract. The fabrics showed 100% antifungal potency against *A. niger*. According to Hamouda et al., 2021 [48], cotton treated with AgNPs had the greatest antibacterial, antifungal, and antiviral activity with 51.7% viral inhibition against MERS-CoV, high antibacterial activity against Gram-positive and Gram-negative bacteria, and the greatest antifungal activity against *A. niger* and *C. albicans*. Chavez et al. [49] also developed cotton fabrics that were antibacterial and antifungal. They used AgNPs to finish the fabric against *E. coli, S. aureus, C. albicans,* and *A. niger*. Fabrics treated with AgNPs showed 100% antibacterial activity and good antifungal activity.

### 3.2. Titanium Dioxide Nanoparticles (TiO_2_NPs)

TiO_2_ is an inorganic material with many applications in textile manufacturing, particularly UV protection [50], self-cleaning, and antimicrobial properties [51]. Due to its unique properties such as stability, non-toxicity, photocatalytic, chemical resistance, and convenient production technique [52], TiO_2_ has drawn a lot of attention. In the presence of TiO_2_, reactive oxygen species (ROS) such as superoxide and hydroxyl radicals can be generated. ROS can damage bacteria’s cell walls, causing them to die. It is this property of TiO_2_ nanoparticles that has been used in antibacterial textiles [53]. Several studies have shown that incorporating TiO_2_ to other metals, metal oxides, polymers, carbon nanoparticles, and matrics enhances the percentage of bacterial killing [54]. Using an in-situ sol-gel approach, Peter et al. [55] investigated how TiO_2_ nanoparticles can be produced and incorporated into cotton fabrics for self-cleaning purposes. The self-cleaning performance of cotton fabrics loaded with TiO_2_ was improved. The pad-dry-cure process was developed by Wang et al. [56] to finish cotton fabric with multifunctional TiO_2_NPs. In a variety of stains, the finished fabric demonstrated excellent self-cleaning properties. A piece of UV-protective cotton fabric was developed by Cheng etal., 2018 [57]. Layer-by-layer self-assembly was used to apply TiO_2_NPs to cotton fabric. The UPF values demonstrated that the nano cotton fabrics provided excellent UV protection and had a good affinity between the nanoparticles and the fabric surface against launderings. 

In 2019, Riaz et al. [58] investigated the applications of TiO_2_ with 3-(Trimethoxysilyl) propyl-*N,N,N*-dimethyloctadecylammonium chloride and 3-(Glycidoxypropyl)trimethoxy-silane in textiles. As a result, they found that treated cotton showed durable super-hydrophobicity, self-cleaning, and antibacterial properties. Alipourmohammadi et al., 2019 [59] reported self-cleaning and antibacterial properties of cotton fabrics with TiO_2_NPs. As compared to uncoated cotton fabrics, TiO_2_NPs-coated materials possess superior self-cleaning and antibacterial properties. Bekraniet al. [60] created antibacterial and UV-protective cotton fabrics coated with TiO_2_NPs. The nano-textiles displayed excellent activity against Gram-negative and Gram-positive bacteria. The UV-blocking of treated samples revealed that when exposed to UV irradiation, all samples have very low transmission.

In 2020, El-Bisiet al. [61] developed cotton fabrics with improved antibacterial and ultraviolet properties after treating them with TiO_2_NPswith Moringa oleifera extract. The UPF and antibacterial properties of TiO_2_NPs cotton fabrics are improved.

The TiO_2_NPs were synthesized by using Aloe vera extract in a green method by Saleem et al. [62]. The TiO_2_-coated fabric demonstrated excellent self-cleaning properties. The tensile strength of the fabric decreased slightly but increased after the TiO_2_ coating. A list of the functionalization of cotton fabrics integrated with TiO_2_NPs is presented in Table 2.

### 3.3. Silica Nanoparticles (SiO_2_NPs)

Silica nanoparticles (SiO_2_NPs) have recently received a lot of attention because of their potential applications in several fields of science and industry. Their properties include self-cleaning, water-repellency, UV protection, and antibacterial properties. Textiles are most modified with nano silica [63]. In cotton fibers, SiO_2_NPs penetrate easily and bind tightly to the fiber structure. Consequently, cellulose hydroxyl groups and SiOH form covalent bonds in SiO_2_NPs. SiO_2_NPs are added to the surfaces of materials to improve their mechanical properties, durability, function, activity, and stability [64].

Rethinam et al. [65] developed antibacterial/ultraviolet cotton fabrics using SiO_2_NPs produced from xerogels at different concentrations (1, 2, and 3% *w*/*v*). Among the different concentrations of SiO_2_NPs used, 3% (*w*/*v*) exhibited better mechanical properties, breaking strength, elongation at break, and tearing strength, and demonstrated the highest antibacterial activity against *S. aureus* and *E. coli,* as well as UV protection. Using SiO_2_NPs, Riaz et al. [66] developed durable superhydrophobicity and antibacterial cotton fabrics. Cotton fabric was treated with SiO_2_NPs using a pad-dry-cure technique.

The results show that the fabric still retains its superhydrophobicity and antibacterial activity even after 20 washing cycles. Additionally, the fabrics comfort properties, like bending rigidity and tensile strength, have improved. According to Zakir et al. [67], SiO_2_NPs were used to fabricate superhydrophobic cotton fabrics. Dip-coating was used to deposit SiO_2_NPs on cotton fabrics. The results showed that cotton sample surface wettability changed from superhydrophilicity to true superhydrophobicity. PFOA-Free Fluoropolymer-Coated SiNPs or Omni Block, created by Kwon et al. [68], demonstrated excellent oil and water repellency on cotton fabrics. PFOA-free fluoropolymer was cross-linked between Si-O-Si groups to produce PFOA-free fluoropolymer-coated SiNPs. After coating the cotton fabric with PFOA-free fluoropolymer-coated SiNPs via a dip-dry-cure method, a rough, high-surface-area oleophobic structure developed. The cotton fabric’s thermal stability and mechanical strength were improved by the coating.

Because SiO_2_NPs have high thermal stability, they can also be used to prepare flame-retardant textiles. In 2021, Shahidi et al. [69] used in-situ synthesis to deposit SiO_2_NPs on cotton fabrics. By impregnating the cotton fabrics with SiO_2_NPs, the flame-retardant properties have greatly improved, and samples have been found to be hydrophilic. Amibo et al. [70] investigated the antibacterial properties of SiO_2_NPs loaded with AgNPs-coated cotton fabrics. Selected strains of bacteria such as *S. aureus*, *E. coli*, and *P. aeruginosa* were tested for antimicrobial activity with improved activities by the treated fabric. Hasabo and Rahma [71] fabricated superhydrophobicity water-repellent cotton fabric coated with SiO_2_NPs and water-repellent agent(WR agent).Water contact angles on the fabric surface of cotton fabrics treated with the WR agent alone remained lower than 20° approximately at the WR agent concentration of 0.3 wt% or less. The hydrophilic surface of cotton fabric was not changed by SiO_2_NPs treatment itself, indicating that water drops were absorbed into fabrics due to the hydroxyl groups on both the cotton and silica NPs surfaces. However, cotton fabrics treated with both silica nanoparticles and the WR agent, a contact angle above 75° can be achieved even at the extremely low WR agent concentration of 0.1 wt%. Therefore, silica nanoparticles and WR agent treatment might be combined to produce superhydrophobicity cotton fabrics. The reported functionalization of cotton fabrics with SiO_2_NPs is presented in Table 3.

### 3.4. Zinc OxideNanoparticles (ZnONPs)

In textile finishing, zinc oxide (ZnO) has gained popularity because of its following numerous advantages: UV protection [50], antibacterial and antifungal properties, and the ability to speed wound healing [72]. ZnO nanoparticles have been deposited or incorporated into cotton using various chemical/physical techniques to develop antibacterial, antifungal, and UV-protective nanotextiles. Table 4. summarizes the functionalization of cotton fabrics treated with ZnONPs.

Using ZnONPs, Fouda et al. [73] fabricated multifunctional medical cotton fabrics. Using secreted proteins from *Aspergillus terreus* AF-1, ZnO nanoparticles were synthesized on cotton fabric to investigate antibacterial activity and UV-protection properties. Bacteria were inhibited by the functionalized fabrics. The ZnONPs have an excellent ability to block UV rays, resulting in an increase in the UPF value of the cotton fabric treated with them. Salat et al. [74] also investigated the antibacterial properties of cotton medical fabrics with ZnONPs and gallic acid (GA). Cotton fabric was uniformly coated with ZnONPs. Despite 60 cycles of washing, the antibacterial efficacy of ZnONPs-GA-coated fabrics remained above 60%. To obtain antibacterial fabrics, Souza et al. [75] used the solochemical process for ZnONPs on cotton fabrics. The antibacterial activity of cotton fabrics against *S. aureus* and *P. aeruginosa* was tested. The antibacterial activity of the treated cotton was higher against *S. aureus* than against *P. aeruginosa*.

In another study, Roy et al. [76] synthesized ZnONPs using a chemical method. ZnONPs were then applied to cotton fabric using dip coating. Antifungal and antibacterial activities of treated samples were examined at various mole concentrations of ZnONPs (1M, 1.5M, 2M, 2.5M, and 3M). The fabrics treated were tested for antifungal activity against *A. niger* as well as antibacterial activity against *S. aureus* and *E. coli*. At a concentration of 2M, the antibacterial and antifungal activity is highest. Mulchandani et al. [77] prepared ZnONPs using a wet chemical method and applied them to cotton fabrics in different concentrations (0.01%, 0.05%, 0.10%, and 0.25%). After 50 cycles of washing, 0.1% of ZnONPs showed excellent antimicrobial activity against *S. aureus* and *K. pneumoniae*. To impart antibacterial activity to cotton (woven, single jersey, rib/double jersey), Momotaz et al. [78] used spin coating and pad-dry-cure methods. The pad-dry-cure technique gave better antibacterial activity than spin coating. Double jersey fabric showed the highest antibacterial activity against (*S. aureus and E. coli*.) than woven and single jersey fabric. In the next study, Mousa and Khairy [79] produced cotton defense clothing. They used a liquid precipitation method to synthesize ZnONPs and investigated the antimicrobial and UV protection of cotton fabrics. ZnONPs were incorporated onto cotton fabrics using the dip and curing method. The nanotreated fabrics showed the highest antimicrobial activity for *S. aureus*, *E.coli*, and *C. albicans,* and the highest UPF values.

Tania and Ali [80] created cotton functional fabrics using the following three different ZnONP recipes: ZnONPs (ZnO-A), ZnONPs with a binder (ZnO-B), and ZnONPs with a binder and wax emulsion (ZnO-C). The treated fabrics were tested within one hour for *S. aureus* and *E. coli*. Nanotreated fabrics significantly reduced the growth of the two bacteria by 50.54–90.43%. ZnO–B and ZnO–C nano fabrics showed 99% reductions. Nano ZnO-B and nano ZnO-C have excellent UPF values. Patil et al. [81] prepared ZnONPs using sono synthesis and applied them to cotton fabrics in 2021. Finished fabrics with ZnONPs have better flexural rigidity because following the deposition of ZnONPs, the stiffness of the cloth increases. An analysis of the cotton fabric’s tensile strength after ZnONPs were deposited that revealed a 5.43% reduction in the tensile strength. On the other hand, the contact angle increased from 38° to 110°. However, the air permeability values after deposition of ZnONPs on cotton fabric decrease approximately by 4.85%. Against *E. coli* and *S. aureus* bacteria, they showed excellent antibacterial activities.

### 3.5. Copper/Copper Oxide Nanoparticles (Cu/CuONPs)

Due to their abundance, availability, and low cost, copper nanoparticles are gaining popularity [82]. As a result, CuONPs are used in a variety of applications, including antifungal, antiviral, antibiotics, anticancer, photocatalytic, biomedical, and agricultural fields [83]. CuONPs possess antimicrobial activity against *Bacillus subtilis*, *E. coli*, *S. aureus*, *Micrococcus luteus, P. aeruginosa, Salmonella enterica, and Enterobacter aerogenes*, as well as antifungal activity against *Fusarium oxysporum* and *Phytophthora capsici*. Accordingly, CuONPs have shown significant antiviral activity against human influenza A (H1N1), avian influenza (H9N2), and many other viruses, including COVID-19 [84].

In 2018, Nourbakhsh and Iranfar [85] prepared cotton fabrics with antibacterial properties by using CuONPs with different concentrations (0.01, 0.03, 0.05, 0.1, 0.2, 0.5, 10%). These fabrics were tested against *E. coli* and *S. aureus* for their antibacterial properties. The antibacterial activity of *E. coli* and *S. aureus* increased with increasing CuONP concentration (99% and 98%, respectively).Based on their results, the optimum concentration of CuONPs was found at 1%. Despite 5 laundering cycles, antibacterial activity for both bacteria decreased by92%. The recovery angle, bending length, and wetting time all increased with increasing CuONP concentrations. A cotton fabric with antibacterial properties was developed by Sun et al. [86] by created an antibacterial cotton fabric by synthesizing CuONPs and applying them to cotton fabrics using atom transfer radical polymerization (ATRP) and electroless deposition(ELD). A uniform distribution of CuONPs was observed on the cotton fabric’s surface. CuONP-functionalized cotton fabric exhibited excellent antibacterial activity against *S. aureus* and *E. coli* even after 30 cycles of washing. CuO nanoparticles were incorporated into cotton fabrics by Paramasivan et al. [87]. Using Cassia alata leaf extract as a reducing agent, CuONPs were synthesized. *E. coli* bacteria were significantly inhibited by nanocotton fabric. Even after 15 washes, these nanocomposites retained antibacterial activity, indicating that the presence of permanent CuNPs in them.

Shaheen et al. [88] treated cotton fabrics with CuONPs to produce antibacterial textiles in 2021. *Aspergillus terreus* AF-1 biomass filtrate was used to synthesize CuONPs. CuO NP-treated cotton fabrics showed significant antibacterial activity against *Bacillus subtilis* and *P. aeruginosa*, but this efficacy was reduced against *S. aureus* and *E. coli.* Alagarasanet et al. [89] also produced a cotton fabric treated with CuONPs for enhanced antibacterial and antifungal properties. Cotton fabrics were coated with CuONPs using the pad-dry-cure technique. They tested the antimicrobial activity against *S. aureus*, *E. coli*, *Pseudomonas fluorescens*, and *B. subtilis*, as well as the antifungal activity against *C. albicans*. Nanocoated fabrics showed better antibacterial and antifungal properties. CuONPs-coated cotton fabrics were also investigated by El-Nahhal et al. [90]. The treated fabric showed improved antimicrobial activity against selected strains of bacteria such as *E. coli* and *S. aureus*. In addition to their antiviral properties, they may also be useful in combating the spread of the COVID-19 Corona Virus. Table 5 summarizes the functionalization of cotton fabrics with Cu/CuONPs.

### 3.6. Gold Nanoparticles (AuNPs)

The optical, electronic, and magnetic properties of AuNPs have drawn a lot of attention in textile research. Textiles also contain AuNPs for electronic and medical applications [91].

In 2018, Shanmugasundaram and Ramkumar [92] attempted to improve the antibacterial property of cotton fabric by coating it with keratin protein and AuNPs using a padded method. AuNPs were synthesized using a chemical reduction method. Incorporating AuNPs and keratin improved antibacterial efficacy against *S. aureus*, *P. aeruginosa*, *E. coli*, and *K. pneumoniae*. A coating of keratin and AuNPs reduced the fabric’s porosity and water absorption. 

Ganesan and Prabu [93] modified cotton fabrics with AuNPs synthesized from chloroauric acid and extract of Acorus calamus rhizome and then applied them to cotton fabrics using pad-dry-cure technology. In addition, the antibacterial activity of treated cotton against *S. aureus* and *E. coli* was excellent. The AuNPs improved the UV-blocking properties of cotton fabric. A study by Baruah et al. [94] focused on improving the catalytic activity of cotton fabrics containing ZnO nanorods and AuNPs. Before AuNPs were deposited on the fabric, ZnONRs were applied. AuNPs were prepared by exsitu synthesis and citrate reduction and applied to a cotton fabric coated with ZnONRs using the dip-coating technique. The photocatalytic dye degradation and recycling properties of the composite materials were excellent. By immersing cotton fabrics in colloidal solutions, Boomi et al. [95] synthesized AuNPs by reducing HAuCl_4_ with *Coleus aromaticus* leaf extract. The antibacterial properties were tested on these fabrics. *Staphylococcus epidermidis* and *E. coli*. A nano cotton fabric was found to have outstanding UV-blocking and antibacterial properties.

Boomi et al. [96] synthesized AuNPs using Croton sparsiflorus leaf extract in 2020 and deposited them on cotton fabric through the pad-dry-cure method to improve their antibacterial, anticancer, and UV properties. Cotton fabrics coated with AuNPs showed excellent antibacterial activity against *S. epidermidis* and *E. coli*, good UPF values, and significant anticancer activity against HepG2. An aqueous extract of Acalypha indica was used by Boomi et al. [97] to prepare AuNPs. A pad-dry-cure procedure was used to coat the intact extract onto the cotton fabric. The antibacterial activity of treated cotton fabric against *S. epidermidis* and *E coli* was evaluated, and it demonstrated remarkable inhibition. Similarly, Dakineni et al. [98] reported that cotton fabrics containing AuNPs were antibacterial, anticancer, and UV protective. Using Pergulariadaemia leaf extract and chloroauric acid, they prepared AuNPs and loaded them on cotton fabrics using pad-dry-cure. Antibacterial activity was significantly enhanced by AuNPs-coated cotton fabric against *S. epidermidis* and *E. coli*, with superior UV-protection efficiency and limited anticancer activity against HepG2. Table 6 summarizes the functionalization of cotton fabrics with AuNPs.

### 3.7. Mixtures of Metal Nanoparticles

To improve the properties of individual MNPs, binary and ternary nanoparticles have been developed and studied. To impart multifunctional properties to cotton fabric, bimetallic nanoparticles (ZnO/TiO_2_NPs) were deposited on the fabric using the sol-gel technique and then applied using the pad-dry-cure method. Nanocomposite cotton fabrics have excellent antimicrobial activity against *E. coli*, high UPF values, and are highly self-cleaning. ZnO and TiO_2_ coatings on cotton fabric can improve multifunctional properties significantly compared to ZnO and TiO_2_ coatings alone [99].

To enhance cotton fabrics’ antibacterial properties, Mamatha et al. [100] used Aloe vera leaf extract to generate Ag/CuNPs. Using aqueous solutions of AgNO_3_ and CuSO_4_.5H_2_O, cotton fabrics infused with Aloe vera leaf extracts were immersed in these metallic source solutions and stirred. Cotton fabrics coated with Ag/CuNPs exhibit good antibacterial activity against *E. coli*, *P. aeruginosa*, *Bacillus cereus*, *K. pneumoniae* and *S. aureus*.

In addition, Rao et al. [101] generated Ag/CuNPs in cotton fabrics using aqueous red sand extracts as a reducing agent. NPs matrices were generated by dipping cotton fabrics in red sander extract solutions. The antibacterial activity of Ag and CuNPs and Ag-Cu bimetallic NPs (BMNPs) was compared. BMNPs generated in cotton fabrics exhibited highly activity against *E. coli, P. aeruginosa, S. aureus, and B. lichinomonas*. Saraswati et al. [102] developed antimicrobial and self-cleaning cotton fabrics using a mixture of Ag/TiO_2_NPs and SiO_2_NPs. Photo-assisted deposition (PAD) method was used to synthesize Ag/TiO_2_NPs. The addition of Tetraethyl orthosilicate (TEOS) as a SiO_2_ precursor to enhance the hydrophilic and self-cleaning properties of TiO_2_ during the modified dip coating process used to impregnate the Ag/TiO_2_ treated cotton fabrics. Due to silica’s structural effects and high dispersion, that demonstrated greater photocatalytic activity. The antimicrobial activity of Ag/TiO_2_ NPs-coated cotton fabrics wastested against *E. coli* bacteria and *C. albicans* fungi. They found that 3% Ag/TiO_2_ has excellent antibacterial and antifungal properties, with a disinfection efficiency of 100%. Due to silica’s structural effects and high dispersion, SiO_2_ coatings demonstrated greater photocatalytic activity than Ag/TiO_2_ coatings alone. Another study coated cotton fabrics with Ag/ZnO and CuNPs to enhance their antibacterial activity, UV protection, and conductivity. For the formation of nanoparticles using functionalized polyethyleneimine (FPEI) or polymethylol (PMC), metal salts such as AgNO_3_, Zn(NO_3_)_2_, and Cu(NO_3_)_2_ were used as precursors. The treated cotton fabrics demonstrated excellent ultraviolet and electrical conductivity, as well as good antibacterial properties even after 20 cycle of washing against *S. ureus* and *E. coli* [103].

In a 2020 report from Ansari et al. [104] Ag, TiO_2_, and ZnO nanoparticles were prepared from AgNO_3_ with trisodium citrate, while TiO_2_NPs were produced by mixing TiCl_4_ and ammonium carbonate. ZnONPs were produced by combining ZnCl_2_ and sodium hydroxide. After immersing cotton fabrics in polyurethane solution, they were immediately immersed in ZnONPs solution and TiO_2_NPs solution. Using the AgNPs solution, the procedure was repeated. The treated fabrics with Ag, ZnO, and TiO_2_NPs showed the best photocatalytic and antibacterial activities against *Shigella*, *Salmonella typhi*, and other bacteria. 

The Gao research group [105] prepared (Ag/ZnO)NPs by chemical precipitation to obtain treated cotton fabrics with improved hydrophobicity, UV resistance, antibacterial, and anti-mildew properties. A cotton fabric was tested for antimicrobial activity against bacteria (*S. aureus, E. coli*) and fungi (*C. albicans*). The antifungal activity of these fabrics was also tested against *Aspergillus flavus*. Silver NPs with anti-mildew properties must contain at least 1% silver, with 3% silver NPs being the best for achieving a proof grade 1 (a proof grade 4 means no mildew resistance). Antibacterial and mildew resistance weredemonstrated by cotton fabrics treated with Ag/ZnO (3% Ag) NPs. Materescu et al. [106] improved the self-cleaning properties of cotton fabrics using commercial aqueous colloidal dispersions of SiO_2_-TiO_2_ nanoparticles (1:0.5; 1:1; 1:1.5). A TiO_2_/SiO_2_NPs mixture enhanced self-cleaning properties, with the highest photocatalytic activity when the molar concentration of TiO_2_/SiO_2_NPs was 1:1.

Silva et al. [107] developed antimicrobial and antiviral cotton fabrics with Ag/TiO_2_NPs synthesized by sonochemistry using AgNO_3_ and trisodium citrate as a reductant and stabilizer. More than 50% of infectious SARS-CoV-2 remains active after prolonged direct contact with self-disinfecting materials that inhibits the proliferation of *E. coli* and *S. aureus*. Table 7 summarizes the functionalization of cotton fabrics with NP mixtures and their applications.

## 4. Conclusions

According to previous studies, the surface modification of cotton fabrics with nanoparticles that provide them with multifunctional properties has been widely studied in the last five years. This has been accomplished using metal and metal oxide nanoparticles (mainly Ag, TiO_2_, SiO_2_, ZnO, CuO, and Au) and mixtures of metal and metal oxide nanoparticles (such as ZnO/TiO_2_, Ag/Cu, Ag/TiO_2_, Ag/ZnO, TiO_2_/SiO_2_, Ag/ZnO/Cu, and Ag/ZnO/TiO_2_). Regarding the synthesis of these nanoparticles, chemical methods are still the most popular. Most of the reducing agents employed in the conventional chemical reduction of metal salts are being replaced by more ecofriendly reductants, such as compounds derived from bacteria, fungi, algae, extracts from various plants. Most cotton coatings are done using immersion (dip-dry) or pad-dry-cure techniques, as well as ultrasonic irradiation. Among the most antimicrobial nanoparticles used over cotton fabrics is silver.TiO_2_ nanoparticles also behave like Ag and exhibit self-cleaning and UV protection properties for textiles.SiO_2_NPs are added to the surfaces of materials to improve their flame-retardant and water repellency properties, in addition to its antibacterial, self-cleaning and UV protection properties. The use of ZnO nanoparticles improves the antibacterial, antifungal and UV protective properties of cotton fabrics. Copper nanoparticles are used in wound healing, and medical applications to give them antibacterial, antifungal and antiviral properties. To achieve the functionalities like antibacterial, anticancer, UV protection, coloration and photocatalysis, AuNPs are used in cotton fabrics. This review may be as an interesting for researchers who want to extend their knowledge of nanotechnology breakthroughs in various applications as household industrial, dressing, wound healing, packaging, footwear, sportswear, protective and medical products. 

## Figures and Tables

**Figure 1 polymers-14-04273-f001:**
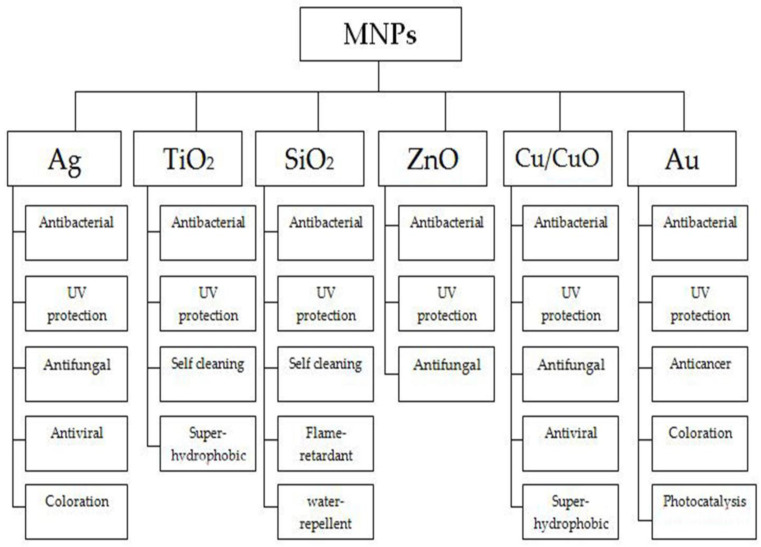
Metal nanoparticles and their functions used in textiles.

**Figure 2 polymers-14-04273-f002:**
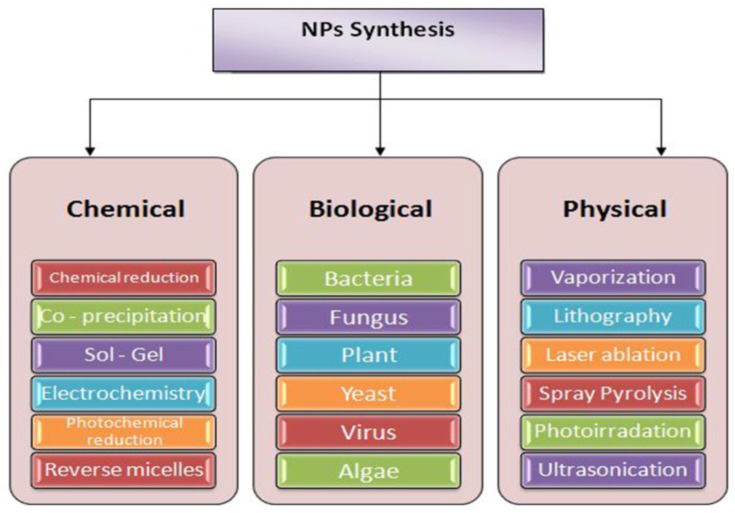
Method of nanoparticles synthesis.

**Table 1 polymers-14-04273-t001:** Summary of the functionalization of cotton fabrics integrated with AgNPs.

Nanomaterials	NPs Size	Synthesis Method	Application Method	Functionality	Ref Year
AgNPs	n.a *	-	Pad-dry-cure	Antibacterial	[41] 2018
AgNPs	n.a *	Seaweed (Padina gymnospora) extract	Pad-dry-cure	Antibacterial and UV protection	[42] 2018
AgNPs	n.a *	Sonochemical	-	Antibacterial	[43] 2019
AgNPs	50–100 nm	Polyol method	Dip coating	Antibacterial and Antifungal	[44] 2019
AgNPs	15–40 nm	Peltophorum pterocarpum leaf extracts	Coating	antimicrobial and wound healing activity	[45] 2020
AgNPs	11.00–83.30 nm	Parkia biglobosa wastewater	Pad-dry-cure	Antibacterial and Antifungal	[46] 2021
AgNPs	91–100 nm	Medicinal plant *Vitex* leaf extract	-	Antibacterial	[47] 2021
AgNPs	n.a *	Chemical method	Coating	Antibacterial, antifungal, and antiviral	[48] 2021
AgNPs	5–20 nm	Chemical method	Exhaustion method	Antibacterial and Antifungal	[49] 2022

* n.a = not available.

**Table 2 polymers-14-04273-t002:** A list of the functionalization of cotton fabrics integrated with TiO_2_NPs.

Nanomaterials	NPs Size	Synthesis Method	Application Method	Functionality	Ref Year
TiO_2_NPs	n.a	In situ sol-gel	Immersion, drying	Self-cleaning	[55] 2018
TiO_2_NPs	n.a	Sol-gel	Pad-dry-cure	Self-cleaning	[56] 2018
TiO_2_NPs	50–120 nm	In situ hydrothermal under sonication	Layer-by-layer self-assembly	UV protection	[57] 2018
TiO_2_NPs	40 nm	Chemical method	Dip coating	Durable super-hydrophobicity, self-cleaning and antibacterial	[58] 2019
TiO_2_NPs	20–25 nm	In situ ultrasonic assisted sol-gel	Immersion, drying, curing	Self-cleaning and antibacterial	[59] 2019
TiO_2_NPs	Less than 50 nm	-	Immersion, heating, drying	Antibacterial and UV protection	[60] 2019
TiO_2_NPs	n.a	-	Immersion, pad-dry-cure	Antibacterial and UV protection	[61] 2020
TiO_2_NPs	11.27 nm	Aloe vera extract in a green method	Pad dry	Self-cleaning	[62] 2021

**Table 3 polymers-14-04273-t003:** A survey of the functionalization of cotton fabrics with SiO_2_NPs.

Nanomaterials	NPs Size	Synthesis Method	Application Method	Functionality	Ref Year
SiO_2_NPs	20–100 nm	Xerogels synthesized from cotton pods	Immersion, drying	Antibacterial and UV protection	[65] 2018
SiO_2_NPs	20–30 nm	-	Pad-dry-cure	Durable superhydrophobic and antibacterial	[66] 2019
SiO_2_NPs	90–150 nm	Stöber method	Dip-coating	Superhydrophobic	[67] 2020
SiO_2_NPs	200 nm	Stöber method	Dip-dry-cure	Oil and water repellency	[68] 2020
SiO_2_NPs	n.a	In-situ sol-gel	Immersion, drying	Flame-retardant	[69] 2021
SiO_2_/AgNPs	n.a	SiO_2_NPs by sol-gel AgNPs by green synthesis	-	Antibacterial	[70] 2021
SiO_2_NPs	150–300 nm	Sol-gel	Immersion, pad-dry-cure	Super hydrophobicity Water-repellent	[71] 2021

**Table 4 polymers-14-04273-t004:** Summary of the functionalization of cotton fabrics treated with ZnONPs.

Nanomaterials	NPs Size	Synthesis Method	Application Method	Functionality	Ref Year
ZnONPs	n.a	(Biological method) secreted proteins by the isolated fungus *Aspergillus terreus* AF-1	Pad-dry-cure	Antibacterial and UV protection	[73] 2018
ZnONPs	<100 nm	In situ sono-chemical	Coating	Antibacterial	[74] 2018
ZnONPs	n.a	Solochemical	Immersion, drying	Antibacterial	[75] 2018
ZnONPs	n.a	Chemical method	Dip coating	Antibacterial and Antifungal	[76] 2020
ZnONPs	n.a	Wet chemical	Pad-dry-cure	Antibacterial	[77] 2020
ZnONPs	n.a	-	Spin coating & Pad-dry-cure	Antibacterial	[78] 2020
ZnONPs	26 nm	liquid precipitation	Dip and curing	Antibacterial, antifungal and UV protection	[79] 2020
ZnONPs	70 (±5) nm	Wet chemical	Mechanical thermo-fixation (Pad-dry-cure)	Antibacterial and UV protection	[80] 2021
ZnONPs	n.a	Sonosynthesis	Coating	Antibacterial	[81] 2021

**Table 5 polymers-14-04273-t005:** Summary of the functionalization of cotton fabrics with Cu/CuONPs.

Nanomaterials	NPs Size	Synthesis Method	Application Method	Functionality	Ref Year
CuNPs	Less than 100 nm	-	Immersion, drying	Antibacterial	[85] 2018
CuNPs	130 ± 20 nm	ATRP and electroless deposition	Immersion, drying	Antibacterial	[86] 2018
CuONPs	40–100 nm	Green synthesis (Cassia alata leaf extract)	Dip coating Shaking+	Antibacterial	[87] 2018
CuONPs	11–47 nm	Green synthesis (Biomass Filtrate of *Aspergillus terreus* AF-1)	Immersion, pad-dry-cure	Antibacterial	[88] 2021
CuONPs	10–100 nm	In situ synthesis	Pad-dry-cure	Antibacterial and Antifungal	[89] 2021
CuONPs	n.a	-	Ultrasonic irradiation	Antibacterial and antiviral	[90] 2021

**Table 6 polymers-14-04273-t006:** Summary of the functionalization of cotton fabrics with AuNPs.

Nanomaterials	NPs Size	Synthesis Method	Application Method	Functionality	Ref Year
AuNPs	8–30 nm Average size 14 nm	Chemical reduction	Padding	Antibacterial	[92] 2018
AuNPs	Less than 100 nm	Green method (extract of Acoruscalamusrhizome)	Pad-dry-cure	Antibacterial and UV protection	[93] 2019
AuNPs	18.5 ± 2.8 nm	Chemical reduction	Dip coating	Photocatalysis	[94] 2019
AuNPs	Different sizes (<20 nm)	Biological reduction	Pad-dry-cure	Antibacterial and UV protection	[95] 2019
AuNPs	16.6–17 nm	Green synthesis	Pad-dry-cure	Antibacterial, anticancer, and UV protection	[96] 2020
AuNPs	19 nm	Green synthesis (Acalypha indica extract)	Pad-dry-cure	Antibacterial	[97] 2020
AuNPs	15–30 nm	Biological reduction (Pergulariadaemia leaves extract)	Pad-dry-cure	Antibacterial, anticancer, and UV protection	[98] 2022

**Table 7 polymers-14-04273-t007:** Summary of the functionalization of cotton fabrics with mixtures of nanoparticles and their applications.

Nanomaterial	NPs Size	Synthesis Method	Application Method	Functionality	Applications	Ref Year
ZnO/TiO_2_NPs	n.a	Sol-gel	Pad-dry-cure	Antimicrobial activity, UV protection, and self-cleaning	Various household industrial and medical applications	[99] 2018
Ag/CuNPs	61 nm	In situ generation using Aloe vera leaf extract	Immersion, drying	Antibacterial activity	Dressing, wound healing, packaging, and medical applications	[100] 2018
Ag/CuNPs	80–90 nm Average size 100 nm	In situ method using aqueous red sand extracts	Dip coating	Antibacterial activity	Antibacterial bed and dressing materials	[101] 2019
Ag/TiO_2_NPs	Anatase 34 nm and rutile 39 nm for Ag/TiO_2_	Photo-assisted deposition (PAD)	Dip coating	Antimicrobial activity and self-cleaning	Footwear application	[102] 2019
Ag/ZnO/CuNPs	FPEI 50 nm PMC 24 nm	Chemical synthesis	Immersion, Pad-dry-cure	Antibacterial activity, UV protection, and conductivity properties	Upholster beds, underwear, and protective clothing	[103] 2019
Ag/ZnO/TiO_2_NPs	Silver colloidal (15.79–97.75 nm) TiO_2_ (9–14 nm) ZnO (13–20 nm)	Chemical synthesis	Immersion	Photocatalytic and antibacterial activities	Hospital and sportswear	[104] 2020
Ag/ZnONPs	Ag 15 nm ZnO 30 nm	Chemical precipitation	Immersion, drying	Hydrophobicity, UV resistance, antibacterial, and anti-mildew activity	Protective clothing	[105] 2020
TiO_2_/SiO_2_NPs	n.a	-	Immersion, pad-dry-cure	Self-cleaning	Self-cleaning textile	[106] 2020
Ag/TiO_2_NPs	1.3 ± 0.4 nm and 27.6 ± 9.9 nm	Sonochemistry method	-	Antimicrobial and antiviral activities	Protective and medical applications	[107] 2021

## Data Availability

The data presented in this study are available on request from the corresponding author.

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
