# Peer review of "Recent Advances in Functionalization of Cotton Fabrics with Nanotechnology"

_polymers, 2022, doi:10.3390/polym14204273_

Round 1

Reviewer 1 Report

The review manuscript summarized recent advances on cotton fabrics functionalization with nanotechnology. Authors majorly stated the metal NP applications in textile fabrics and their functionalization including antimicrobial (antibacterial, antifungal), antiviral, UV protection, self-cleaning, water-repellent, and flame-retardant properties. This review is useful to study on imparting cotton fabrics functionalization by nanotechnology.

Some concerns:

1.     In title, only cotton fabrics was presented. In fact, the lycra in Table 3 and polyester in Table 6 were referred to. Please modified them.

2.     In Tables, the first column “Types of Fabric” is most Cotton. Therefore, this column should be deleted.

3.     Throughout manuscript, the microbiology Latin names in written should be carefully revised. For example, “Escherichia coli and staphylococcus aureus” changed into “E. coli and S. aureus” excluding the first presence in manuscript.

Reviewer 2 Report

The manuscript   is a rewiev of literature about functionalization of cotton fabrics with  nanotechnology . References are well selected, they reflect the works that has been carried out in the world in recent years. The manuscript is written correct except  few mistakes, especially in the names of the bacteria.

This manuscript seems to be useful to the readers working on functionalization of cotton fabrics. I think the manuscript is suitable for its publication in Materials.

Reviewer 3 Report

The paper is quite well written and easy to read. Nevertheless some major improvements are neccessary:

1) Section 2 (page 2) should be  redrafted or at least a reference given at the end of the first sentence. The types of nanotechnology-produced materials in this section are the same as it was decreibed in the reference [21]. Please check the descriptions in this chapter carefully so that the content does not raise any doubts or associations with other uncited work.

2) Check carrefully the references if names of the authors are in correct form (i. e. ref.no 29 probably there are the full names and the first letter of the surname is used)

3) Citation at page 11 of the reference [89} - there is a misteke in the Author's name

Reviewer 4 Report

The review entitled "Recent advances in functionalization of cotton fabrics with nanotechnology", submitted by Tarek Abou Elmaaty  et al, is definitely a potential candidate for publishing in Polymers.

However, not in its present form!

The manuscript contains too many typos and elliptic/confusing/clumsy formulations. I've tried to signal these shortcomings in the attached, reviewed form of the manuscript that contains over 70 comments/observations/suggestions. And there may be more instances needing corrections that I missed. 

The authors are encouraged to carefully revise the whole text for English spelling, grammar and phrasing. This thorough action would probably result in a publishable version.

Round 2

Reviewer 3 Report

Thank you for the corrections. I accept this paper in present form.

Author Response

thank you for the reviewer comments

Reviewer 4 Report

The revised form of the manuscript is a consistently improved version of the previous one. Most of the observations, suggestions, signaled shortcomings have been addressed. There are some "leftover" typos that I've been able to "catch" in the attached reviewed form of the manuscript.

Author Response

The typo mistakes highlighted by the reviewer have been corrected